# Moral Cultivation and Divine Revelation: James Legge's Religious Interpretation of the *Yijing* (Book of Changes)

**John T. P. Lai**

Department of Cultural and Religious Studies, The Chinese University of Hong Kong, Hong Kong, China; johntpl@cuhk.edu.hk

**Abstract:** James Legge (1815–1897), arguably the most prominent missionary sinologist in the nineteenth century and the founding Professor of Chinese in Oxford in 1876, produced an English translation of the *Yijing* (Book of Changes), the prominent Chinese classic, in 1882. This translation was included in Max Müller's monumental *Sacred Books of the East* series. While existing scholarship has outlined some background and features of Legge's *Yijing* translation, this version deserves more in-depth textual analysis to unearth Legge's primary sources of reference and theological positions behind his interpretive approach. Perceiving the *Yijing* as a Confucian classic with profound moralistic connotations, Legge even revered it as a "sacred book" containing certain elements of divine revelation. He asserted that the Chinese term *Shangdi* (Supreme Ruler) referred to the Christian God, insisting that "God" was the "correct" translation of *Shangdi*, and that the operations of nature in the different seasons are the work of *Shangdi*. This paper examines Legge's pioneering attempt of translating the *Yijing* to the West, with special reference to his religious interpretation of the seminal Chinese classic. This endeavor engendered profound inter-religious encounters and dialogues between Confucianism and Christianity.

**Keywords:** James Legge; *Yijing* (Book of Changes); cross-cultural translation; religious interpretation

## 1. Introduction

The multifarious symbolism and religious connotation of the sixty-four hexagrams in the *Yijing* (The Book of Changes), the highly venerated and influential Chinese classic (Smith 2008), captured considerable attention from leading Christian missionaries in late imperial China. James Legge (1815–1897), arguably the most prominent missionary sinologist in the nineteenth century and the founding Professor of Chinese in Oxford in 1876, produced an English translation of the *Yijing* in 1882. This translation was incorporated into Max Müller's monumental *Sacred Books of the East* series (Legge 1882a).[1] Norman Girardot has offered some fascinating insights on the life and work of Legge, who emerged as an eminent nineteenth-century cross-cultural pilgrim, missionary sinologist, and translator of Chinese classics (Girardot 2002). While existing scholarship has mapped out the general background and features of Legge's translation/interpretation of the *Yijing*,[2] Legge's version deserves more exhaustive research and critical textual analysis to unearth his major sources of reference and theological views behind his interpretive approach. While Legge perceived the *Yijing* as a "historical document" (Hon 2005, pp. 320–26) or "book of history" (Wu and Hon 2018, pp. 7–9) on the socio-political system in the early Zhou Dynasty, the text was fundamentally revered as a Confucian classic with profound moralistic connotations, or even a "sacred book" containing some facets of divine revelation. This paper examines Legge's pioneering attempt of introducing and translating the *Yijing* to the West, with special reference to his religious interpretation of this Chinese classic from the perspectives of moral cultivation and divine revelation.

## 2. *Yijing* for Moral Cultivation: References to the *Daily Lectures*

Acknowledging the fact of not having the "help of able native scholars", Legge remarked that the *Rijiang Yijing jieyi* (日講易經解義 *The Daily Lectures on Elucidating The Book of Changes*, hereafter *Daily Lectures*)[3] emerged as a major source of reference for his interpretation of the *Yijing* in the process of translation.[4] Authorized by the Kangxi Emperor, *Rijiang Yijing jieyi*, a work in the imperial series of "Daily Lectures" on the Chinese classics, was edited by Niu Niu 牛鈕 and Sun Zaifeng 孫在豐 and completed in the twenty-second year of the Kangxi reign (1683). With a foreword by the Kangxi Emperor, the work was published, widely disseminated, and emerged as an important reference work for the emperor's study of the *Yijing* (Yang 2008, pp. 93–137; 2012, pp. 325–75). One principal theme of the *Daily Lectures* constitutes "moral cultivation" along the line of the Confucian role ethics, particularly the lofty ideal of becoming a *junzi* (君子 superior man) and sage/sage king (Ames 2021).

Legge professed the importance of *Daily Lectures* in shaping his views: "Any sinologist who will examine [the *Daily Lectures*]… will see the agreement between my views and those underlying its paraphrase." (Legge 1882a, p. xiv) While Richard Rutt noted that Legge made much use of *Daily Lectures*,[5] no systematic research has hitherto been undertaken to investigate the significance of *Daily Lectures* on Legge's interpretation of the *Yijing*. Filling this research gap, this section focuses on the examination of the influence of *Daily Lectures* on Legge's moral interpretation of the *Yijing*.[6] A thorough analysis of this invaluable primary text will help to trace the sources of Legge's understanding of the classic, the intertextual connection with the Chinese *Yijing* commentaries, as well as his theological position towards the Confucian moral principles.

*Increasing the "Heavenly Principle" and Diminishing the "Human Desires"*

Regarding the impact of *Daily Lectures* on Legge's interpretation, a number of hexagrams foreground the themes of moral cultivation, self-cultivation, as revered in the Confucian tradition. A case in point is Legge's commentary on the Hexagram *Jing* (井 The Well ䷯)[7], which stated that "the 'Daily Lecture' observes here:—'The cultivation of one's self, which is represented here, is fundamental to the government of others.'" (Legge 1882a, p. 328) More remarkably, Legge adopted a distinctive approach in interpreting the moral/philosophical underpinnings of the Hexagram *Qian* (謙 Modesty ䷎). The entire handwritten marginal note found in the *Qian* hexagram in his personal copy of the *Daily Lectures* was translated as follows:

> "The five yin lines above and below symbolize the earth; the one yang line in the centre is "the mountain in the midst of the earth." The many yin lines represent men's desires; the one yang line, heavenly principle. The superior man, looking at this symbolism, diminishes the multitude of human desires within him, and increases the single shoot of heavenly principle; so does he become grandly just, and can deal with all things evenly according to the nature of each. In whatever circumstances or place he is, he will do what is right."[8]

Legge explicitly affirmed that the single *yang* line in the third place of the *Qian* hexagram symbolizes the "heavenly principle" (*tianli* 天理), while the remaining *yin* lines representing the "human desires" (*renyu* 人欲). The idea of "[diminishing] what is excessive" and "[increasing] where there is any defect" in the Great Symbolism [*Da xiang zhuan* 大象傳] was interpreted as diminishing the "human desires" and increasing the "heavenly principle".

Along the line of the aforementioned Hexagram *Qian*, it is worth noting that Legge also underscored the theme of "heavenly principle" and "human desires" in his interpretation of the Hexagram *Guai* (夬 Break-through ䷪): "If but a single small man be left, he is sufficient to make the superior man anxious; if but a single inordinate desire be left in the mind, that is sufficient to disturb the harmony of heavenly principles. The eradication in both cases must be complete, before the labour is ended." (Legge 1882a, p. 249 note)

As Thomas Selover maintained, "heavenly principle" is the natural pattern of things and of the heart/mind (*xin* 心). According to the Neo-Confucians, when "human desires", or "selfish desire", increase within the heart/mind, "heavenly principle" is diminished and vice versa; the perfectionist goal was complete elimination of "selfish desire." (Yao 2003, pp. 616–17) Legge unhesitatingly referred to the leading Neo-Confucians of the Song Dynasty, particularly Cheng Yi (程頤, 1033–1107) to buttress his argument along this line of interpretation: "Khăng-jze [Cheng Yi] says that 'the ordinances of Heaven' are simply the natural and practical outcome of 'heavenly principle;'—in this case what should and may be done according to the conditions and requirements of the time" (Legge 1882a, p. 251 note); and "'If a man,' says Khăng-jze [Cheng Yi] 'cherish a single illicit desire in his mind, he has left the right way.'" (ibid., p. 321 note)

In illustrating the prominence of "heavenly principle" in the Hexagram *Qian*, Legge's interpretation of the Hexagram *Gen* (艮 Keeping Still/Mountain ䷳) should be taken into account because the trigram *Gen* constitutes the lower trigram of the Hexagram *Qian*. Legge maintained that the trigram *Gen* "denotes the mental characteristic of resting in what is right; especially resting, as it is expressed by Chinese critics, 'in principle,'—that which is right, on the widest scale, and in the absolute conception of the mind; and that which is right in every different position in which a man can be placed. We find this treated of in the *Great Learning* [*Da xue* 大學] (Commentary, chapter 3), and in the *Doctrine of the Mean* [*Zhong yong* 中庸], chapter 14… The back alone has nothing to do with anything beyond itself—hardly with itself even; all that it has to do is to stand straight and strong. So should it be with us, resting in principle, free from the intrusion of selfish thoughts and external objects."[9] In his commentary, the dominant theme of "resting in principle" has been reiterated, and "principle" here should be referring to "heavenly principle" in contrast to "selfish thoughts and external objects". In his interpretation of the Hexagram *Gen*, Legge, albeit not directly mentioning *Daily Lectures*, should have consulted *Daily Lectures* with his references to other Confucian texts, particularly the *Great Learning* and the *Doctrine of the Mean*. As a matter of fact, *Daily Lectures* distinctly alludes to *Great Learning* chapter three[10] and *Doctrine of the Mean* chapter fourteen[11] in elaborating the notion "resting in principle".[12]

On the Confucian understanding of the moral nature of men, Legge argued that "Confucianism teaches men the discharge of their duties in the various relations of life. It regards the moral nature as conferred on men by God." (Legge 1884, p. 2) By quoting *The Doctrine of the Mean*, "What heaven has conferred is called THE NATURE; an accordance with the nature is called THE PATH OF DUTY", Legge contended that "while man is by his moral nature constituted a law to himself, he is so by the act and decree of God; a religious sanction is given to all his relationships and his performance of their duties." (ibid., p. 14) Regarding the notion of "heaven", Legge believed that "man's existence, nature, and duties are from a Supreme Being, now indicated by the impersonal term Heaven, and now called by the personal name of Supreme Ruler, equivalent to our designation God." (Legge n.d., p. 567) Furthermore, the Being who is called "Heaven" in the Chinese classics is "deified heaven" and "The visible heaven, deified, is the chief god of the Chinese." (Legge 1880a, p. 17) All in all, in Legge's view, Confucianism is not merely a system of morality, but also a religion. He asserted that "in this highest of the moralities of Confucianism there is also the element of religion… From time immemorial, there has been in China the belief of one Supreme Being, first indicated by the name heaven, and then by the personal designation of God as the Supreme Lord and Ruler." (Legge 1884, p. 16) In other words, since the moral nature of men was conferred by Heaven/God, men's moral cultivation by increasing the "Heavenly Principle" has a profound religious dimension and connotation.

### 3. *Yijing* with Divine Revelation: Primitive Monotheism in Ancient China

Regarding the distinctive religious dimension of the practice of divination connected with the *Yijing*, Legge explicitly expressed a suspicious and pejorative attitude: "Of course all divination is vain, nor is the method of the Yi less absurd than any other." (Legge 1882a,

p. 43) On the other hand, Legge believed that the ancient Chinese were monotheists who had received some divine revelation as documented in the Chinese classics. Legge contended that there was "a time when the religion of China was a pure monotheism," (Legge 1877, p. 4) and that "a primitive monotheism in China is more in accordance with the testimony of the Bible than any other, and that the usage of Thien and Tî, all along the course of history, struggling against the corruptions of that primitive monotheism." (Legge 1880a, p. 22) Concerning the use of religious terms, Legge agreed with Matteo Ricci's (1552–1610) affirmative view to the central question: "Did the Chinese really mean GOD when they spoke of T'ien (Heaven) and Shang Ti (the Supreme Ruler)?" (Legge 1888, p. 58) In other words, Legge regarded the term *Shangdi* (上帝 the Supreme Ruler) in the ancient Chinese classics as the equivalent of the Judeo-Christian God, and trusted that the Chinese people were capable of receiving the Christian faith through their extant terminology, that monotheism was not totally alien to the Chinese, and that the truth of God had been revealed to them at some point. Furthermore, "This monotheistic faith was at the foundation of the doctrine of the Literati, enunciated from the earliest time, and ruling it down to the present day." (Legge n.d., p. 567).

In summarizing the relation between Confucianism and Christianity, Legge stipulated that "the *Ti* (帝) and *Shang-ti* (上帝) of the Chinese classics is God–our God–the true God." (Legge 1877, p. 3) In his translations of the Confucian classics, Legge was also a strong proponent of the use of "God" for rendering *Shangdi*. He declared that "I have spoken of the Chinese terms Ti and Shang Ti, and shown how I felt it necessary to continue to render them by our word God, as I had done in all my translations of the Chinese classics since 1861." (Legge 1882a, p. xix) When translating the *Yijing*, Legge contended that "'God' is really the correct translation in English of Ti," (ibid., p. 51) and that "I came to the conclusions that Ti, on its first employment by the Chinese fathers, was intended to express the same concept which our fathers expressed by God… when I render Ti by God and Shang Ti by the Supreme God, or, for the sake of brevity, simply by God." (ibid., p. xx) Legge was pleased to pronounce that "a great majority of the Protestant missionaries in China use Ti and Shang Ti as the nearest analogue for God."(ibid.) In this connection, the role of Christian missionaries, for Legge, was to "quicken them [the Chinese people] to the recognition" of the self-existence of the monotheistic God *Shangdi*. (Legge 1880a, p. 20)[13]

### 3.1. Operations of God in Nature

Along this line of Legge's interpretation, the monotheistic *Shangdi* revealed his operations in nature as illustrated in the *Yijing*. In the "Treatise of Remarks on the Trigrams" (*Shuo gua zhuan* 說卦傳), Legge elaborated his arguments on "King Wan's Scheme of the Trigrams" (*Wenwang bagua* 文王八卦) from the Christian perspective of "operations of God in nature", with special reference to the paragraph 8 "God comes forth in Zan [Zhen 震 ☳] (to his producing work); He brings (His processes) into full and equal action in Sun [Xun 巽 ☴]; they are manifested to one another in Li [離 ☲]; the greatest service is done for Him in Khwan [Kun 坤 ☷]; He rejoices in Tui [Dui 兌☱]; He struggles in Khien [Qian 乾 ☰]; He is comforted and enters into rest in Khan [Kan 坎 ☵]; and he completes (the work of) the year in Kan [Gen 艮 ☶]." (Legge 1882a, p. 425) Legge further remarked that this paragraph "sets forth the operations of nature in the various seasons, as being really the operations of God, who is named Ti [帝], 'the Lord and Ruler of Heaven.' Those operations are represented in the progress by the seasons of the year, as denoted by the trigrams." (ibid., p. 426) King Wen, Legge argued, "might set forth vividly his ideas about the springing, growth, and maturity in the vegetable kingdom from the labours of spring to the cessation from toil in winter. The marvel is that in doing this he brings God upon the scene, and makes Him in the various processes of nature the 'all and in all.'"[14] Legge maintained that this *Yijing* paragraph "speaks directly of God," and the following paragraph "speaks of all things following Him, from spring to winter, from the east to the north, in His progress throughout the year." (Legge 1882a, p. 52)



To elucidate his argument, Legge was quoting a Chinese *Yijing* scholar Wan Jing (萬經), of the Kangxi period, in his work entitled *A New Digest of Collected Explanations of the Yijing* (新輯易經集解), saying, "God (Himself) cannot be seen; we see Him in the things (which He produces)."[15] In this connection, Legge further asserted that Wan Jing's words are strikingly similar to those of the apostle Paul in his Epistle to the Romans. (Legge 1882a, p. 52) Here Legge was referring to Romans 1:20, "For the invisible things of him from the creation of the world are clearly seen, being understood by the things that are made, even his eternal power and Godhead; so that they are without excuse." All in all, Legge argued that the author of the "Treatise of Remarks on the Trigrams" gives "emphatic testimony to God as renewing the face of the earth in spring, and not resting till He has crowned the year with His goodness."[16] Once again Legge was attempting to place the *Yijing* in juxtaposition with the Bible by alluding to the biblical passage, "Thou crownest the year with thy goodness; and thy paths drop fatness." (Psalms 65: 11).

Apart from making frequent references to the Bible to illustrate some *Yijing* concepts, Legge went a step further by quoting from the source of Christian hymns. He drew a comparison between the operations of God in four seasons as depicted in "Treatise of Remarks on the Trigrams" and the famous hymn on the Seasons by the Scottish poet James Thomson (1700–1748).[17] Legge highlighted that "The first time I read these paragraphs with some understanding, I thought of [James] Thomson's Hymn on the Seasons, and I have thought of it in connexion with them a hundred times since." (Legge 1882a, p. 52)

> These, as they change, Almighty Father, these
> Are but the varied God. The rolling year
> Is full of Thee. Forth in the pleasing **spring**
> Thy beauty walks, Thy tenderness and love.
> Then comes Thy glory in the **summer** months,
> With light and heat refulgent. Then Thy sun
> Shoots full perfection through the swelling year.
> Thy bounty shines in **autumn** unconfined,
> And spreads a common feast for all that lives.
> In **winter** awful Thou![18] [emphasis added]

### 3.2. The Work of the Spirit of God

On the operations of God in nature, Legge went further to underline the work of the Spirit (*Shen* 神) of God. He stated that the author of "Treatise of Remarks on the Trigrams" speaks of the work of God in nature in all the year as a progress through the trigrams, and as being "effected by His Spirit." (Legge 1882a, p. 50) Specifically in his translation and interpretation of paragraph 10 of "Treatise of Remarks on the Trigrams", "When we speak of Spirit we mean the subtle (presence and operation of God) with all things [神也者，妙萬物而言者也]." (ibid., p. 427) To substantiate this claim, Legge was quoting the Chinese scholar Liang Yin (梁寅, 1309–90), of the Ming Dynasty, who commented on this passage "The spirit here simply means God. God is the personality (literally, the body or substantiality) of the Spirit; the Spirit is God in operation. He who is lord over and rules all things is God; the subtle presence and operation of God with all things is by His Spirit."[19] Regarding the interpretation of Shen [神], Legge argued that "The language is in fine accord with the definition of shan [神] or spirit," (Legge 1882a, p. 53) given in "The Great Appendix", Section I, paragraph 32 (繫辭上傳第五章), "That which is unfathomable in (the movement of) the inactive and active operations is (the presence of a) spiritual (power) [陰陽不測之謂神]." (ibid., p. 357) According to Legge, the best commentary on the inscrutable and unfathomable characteristic of the spiritual operations of God is supplied by paragraphs 8 to 10 of "Treatise of Remarks on the Trigrams". He maintained that "The 'Spirit' is that of 'God'; and this settles the meaning of tao [道], the defined course of things, in paragraph 24,[20] as being the course of nature."[21] Once again Legge alluded to the Bible to illustrate this course of nature: "God worketh all in all."[22]

In comparison to the definition of Shen or "Spirit" in "The Great Appendix", Section I, paragraph 32 [陰陽不測之謂神] and the doctrine of the agency of God in "Treatise of Remarks on the Trigrams", Legge commented on the "Treatise on the Thwan, or king Wan's Explanation of the entire Hexagrams" (*Tuan zhuan* 彖傳) of Hexagram *Guan* (觀卦 Contemplation/View ䷓): "When we contemplate the spirit-like way of Heaven, we see how the four seasons proceed without error. The sages, in accordance with (this) spirit-like way, laid down their instructions, and all under heaven yield submission to them [觀天之神道，而四時不忒。聖人以神道設教，而天下服矣]." (Legge 1882a, p. 230) "The spirit-like way of Heaven" is, in Legge's view, "the invisible and unfathomable agency ever operating by general laws, and with invariable regularity, in what we call nature."(ibid.) In this light, Legge's proposition falls in line with the theological thought of General/Natural revelation, which refers to a revelation being universally available, in the sense that the natural world, including human nature, that is available to all may reveal the work of God. (Wahlberg 2020)

### 3.3. Presenting Offerings to God

Apart from demonstrating the work and operation of God in the *Yijing*, the presentation of offerings to God was also a particularly noteworthy theme in Legge's translation. Legge acknowledged the worship of God in Confucianism: "there is in Confucianism a worship of God Himself… For between three and four thousand years at the least, there has been the worship of this Being; but as formally approved and organized by the ordinances of the State… The greatest occasion of the imperial religious celebration is at the earliest dawn on the morning of the winter solstice at 'the Altar of Heaven'." (Legge 1884, pp. 16–17) The term "God" and presenting offerings to God by the sages and ancient kings feature prominently in Legge's translation, such as that for the Hexagram *Yi* (益 Increase ䷩) on the second line: "Let the king, (having the virtues thus distinguished), employ them in presenting his offerings to God (王用享于帝), and there will be good fortune." (Legge 1882a, p. 150) Translating the Hexagram *Yu* (豫 Enthusiasm ䷏), Legge highlighted the importance of the use of music at sacrifices as assisting the union produced by those services between God and his worshippers: "The ancient kings, in accordance with this, composed their music and did honour to virtue, presenting it especially and most grandly to God (先王以作樂崇德，殷薦之上帝), when they associated with Him [at the service] their highest ancestor and their father." (ibid., pp. 287–88) In similar fashion, the Hexagram *Huan* (渙 Dispersion ䷺) touches upon the presentation of offerings to God: "The ancient kings, in accordance with this, presented offerings to God (先王以享于帝), and established the ancestral temple." (ibid., p. 341) Legge discussed the image of "wind moving above the water" as referring to the "idea of dissipation in the alienation of men from the Supreme Power…; a condition which the wisdom of the ancient kings saw could best be met by the influences of religion." (ibid., p. 37)

Another notable example on presenting offerings to God can be found in the Hexagram *Ding* (鼎 Cauldron ䷱). As an ancient cooking vessel and symbol for imperial power, the cauldron's sacrificial function was further highlighted: "The sages cooked their offerings in order to present them to God (聖人亨以享上帝), and made great feasts to nourish their wise and able (ministers)." (ibid., p. 255) While affirming "'God' is here Shang Ti", Legge took this occasion to criticize Thomas McClatchie's *Yijing* translation and mythological interpretation, with *Shangdi* being a "pagan" god, by remarking that "Canon McClatchie translates 'the First Emperor,' adding in a note, 'The Chinese Jupiter, the Emperor of gods and men!'"[23] All in all, Legge's insistence of the use of "God" for translating "*Shangdi*" and his unrelenting criticism of McClatchie's translation should be examined against the context of intense debates among Protestant missionaries over the "Term Question", chiefly revolving around the translation of "God" into Chinese, in the late nineteenth and early twentieth centuries.[24]

## 4. Concluding Remarks

James Legge's stupendous enterprise of translating the Chinese classics, including the *Yijing*, was part and parcel of his "oriental pilgrimage", as Norman Girardot has aptly put it (Girardot 2002). In this paper, the discovery and discussion of the *Daily Lectures* has thrown new light on Legge's major source of Chinese *Yijing* commentaries for interpreting the *Yijing*, as well as his theological position towards the religiosity of Confucian moral principles and cultivation. Despite the fact that Legge's *Yijing* drew upon the Latin version produced by a group of Anti-Figurist Jesuits in the early nineteenth century,[25] his interpretation exhibited some traces of the Jesuit Figurists, like Joachim Bouvet (1656–1730) and their Christianized *Yijing*,[26] by claiming that the Chinese term *Shangdi* meant "God–our God–the true God," insisting that "God" was the "correct" translation of *Shangdi*, and that the operations of nature in the various seasons, as denoted by the eight trigrams, constitute the work and natural revelation of *Shangdi*/Spirit of God. While the *Yijing* could be viewed as a kind of historical document of the Zhou Dynasty, Legge fundamentally perceived the *Yijing* to be a "sacred book" of China containing distinctive Confucian moral principles, and more importantly, the divine revelation of *Shangdi*, the Chinese name for the Christian God. In this light, Richard Smith's assertion that Legge had "no love of China and no respect for the *Yijing*" (Smith 2012, p. 107) is worthy of serious reconsideration.

Notwithstanding his own missionary agenda and ideological bias, Legge's provocative reading, albeit deviating from traditional Chinese interpretations, has expanded the possibility of diversified interpretations of this ancient Chinese classic imbued with multifarious layers of meaning and profuse symbolisms. Furthermore, Legge's pioneering endeavor of rendering the *Yijing* for English-speaking audiences not only demonstrates developments and trends in nineteenth-century European sinology and the emergence of comparative study of religions, but also engendered extensive cross-cultural encounters and profound inter-religious dialogues between Christianity and Confucianism.

On the basis of the current new perspective of Legge's translation, further efforts to bring other contemporary Protestant translations of the *Yijing*, particularly that of Thomas McClatchie (1876), which conveys a divergent interpretation, together in a comparative framework will offer a deeper understanding of the complexity and creativity of the Victorian *Yijing* translations, which were deep in theology and broad in their global reach and networks.

**Funding:** This paper was funded by the General Research Fund from the Research Grants Council of Hong Kong (Project no. CUHK 14611820: "Protestant Interpretations of the Yijing: A Comparative Study of Thomas McClatchie and James Legge in Late Qing China").

**Institutional Review Board Statement:** Not applicable.

**Informed Consent Statement:** Not applicable.

**Data Availability Statement:** The data presented in this study are available on request from the corresponding author. The data are not publicly available due to copyright issues.

**Conflicts of Interest:** The author declares no conflict of interest.

## Notes

[1] The *Sacred Books of the East* series is a project against the broader context of Müller's comparative study of religions and the science of religion. See Müller (1873).

[2] See, for example, Shchutskii (1980, pp. 28–35); Girardot (2002, pp. 273–75, 365–74); J. Smith (2012, pp. 107–8).

[3] *Rijiang Yijing jieyi* is translated as *The Daily Lectures on the Book of Changes* in Sophie Ling-chia Wei (2020), p. 80.

[4] "It would be tedious to mention the many critical editions and commentaries that I have used in preparing the translation. I have not had the help of able native scholars... The want of this, however, has been more than compensated in some respects by my copy of the 'Daily Lectures on the Yi,'... The friend who purchased it for me five years ago [1877] in Canton was obliged to content himself with a second-hand copy; but I found that the previous owner had been a ripe scholar who freely used his pencil in pursuing his studies. It was possible, from his punctuation, interlineations, and many marginal notes, to follow the

exercises of his mind, patiently pursuing his search for the meaning of the most difficult passages. I am under great obligations to him." See Legge (1882a, pp. xx–xxi).

5    Richard Rutt noted that Legge made much use of *Yuzhi rijiang Zhouyi* [sic] *jieyi*, but without going into further detail. See Richard Rutt (2002, p. 69).

6    Legge's personal library of Chinese classics, including his copy of *Daily Lectures* (Niu and Sun 1683) filled with some marginal notes and commentary, was deposited in the Rare Book Division, New York Public Library (*OVQ 91-12978).

7    The English translation of the hexagrams is taken from Wilhelm (1967).

8    James Legge, *The Yi King*, p. 287 note. The original marginal note in Chinese reads "上下五陰，地之象也。一陽居中，地中有山也。五陰之多，人欲也。一陽之寡，天理也。君子观此象，哀其人欲之多，益其天理之寡，則廓然大公，自可以稱物平施，無所處 而不當也。" See *Rijiang Yijing jieyi*, vol. 5, p. 5. (New York Public Library copy).

9    James Legge, *The Yi King*, p. 177 note. The phrase "the intrusion of selfish thoughts and external objects" should be paraphrased from the lines from *Daily Lectures*, "蔽於己私，外奪於物欲" (vol. 12, p. 14).

10    *Daily Lectures* refers to *Great Learning* chapter 3 to elaborate the philosophical significance of "resting in principle" of the Hexagram *Gen*, "又不可以不思大學言止仁。止敬。止慈。止孝。止信。" See *Rijiang Yijing jieyi*, vol. 12, p. 18. 5. For Legge's translation of *Great Learning* chapter 3, see Legge (1861, vol. I, p. 226).

11    Echoing the Great Symbolism of the Hexagram *Gen* (兼山艮，君子以思不出其位), Legge quoted *Doctrine of the Mean* chapter 14 (君子素其位而行; The superior man does what is proper to the station in which he is; he does not desire to go beyond this.) by referring to the commentary of *Daily Lectures*. See Legge (1861, vol. I, pp. 259–60).

12    "We find this treated of in the Great Learning (Commentary, chapter 3), and in the Doctrine of the Mean, chapter 14, and other places." Legge (1882a, p. 177).

13    For Legge's broader views on the Chinese religions, particularly Confucianism and Taoism, see Legge (1880b).

14    James Legge, *The Yi King*, p. 51. Legge's use of the phrase "all and in all" is an allusion to the biblical passage "Where there is neither Greek nor Jew, circumcision nor uncircumcision, Barbarian, Scythian, bond nor free: but Christ is all, and in all." (Colossians 3:11).

15    James Legge, *The Yi King*, p. 52. The original commentary reads "帝不可見，即物見之". See Wang (1686, vol. 4, p. 3). (New York Public Library copy, *OVQ+ 09-266).

16    James Legge, *The Yi King*, p. 52. Legge also maintained that this particular passage refers to the "creation of the world": "In the chapter Choue-koua 說卦 (5) we read these words—'The *Ti* 帝 or the Lord began to go out by the east.' The text makes use of the word Tching 震, which is one of the eight radical symbols of the Y-king, and which designates east and west. It afterwards goes through the seven others, and finishes with Ken 艮, which denotes a mountain. The majority of interpreters agree that the subject here discussed, is the creation of the world." See Legge (1852, p. 66).

17    James Thomson (1700–1748), a Scottish poet and playwright, was known for his poems *The Seasons* and *The Castle of Indolence*, and for the lyrics of "Rule, Britannia!".

18    James Legge, *The Yi King*, p. 52. For the entire hymn, see Thomson (1792, pp. 236–40).

19    James Legge, *The Yi King*, p. 53. The original passage of Liang Yin reads "神，即帝也。帝者神之體，神者帝之用。故主宰萬物者，帝也。所以『妙萬物』者，『帝』之神也。" See Li (1715, vol. 17, pp. 15–16).

20    "The successive movement of the inactive and active operations constitutes what is called the course (of things)." [一陰一陽之謂道] See Legge (1882a, p. 355).

21    Ibid., p. 358.

22    See 1 Corinthians 12:6: "And there are diversities of operations, but it is the same God which worketh all in all." (King James Version).

23    Ibid. For McClatchie's mythological interpretation of the *Yijing*, see Lai (2021a, pp. 107–21).

24    See, for example, Eber (1999, pp. 135–61).

25    Jean-Baptiste Régis and Julius von Mohl, *Y-King: Antiquissimus Sinarum Liber* (*Yijing*: The Oldest Chinese Book) (Stuttgart & Tübingen: J. G. Cottae, 1834–1837). Legge referred to this Latin translation on many occasions. For instance, he remarked that "The translation of Regis and his coadjutors is indeed capable of improvement; but their work as a whole, and especially the prolegomena, dissertations, and notes, supply a mass of correct and valuable information." See Legge (1882a, p. 9; 1882b, p. 402).

26    For the discussion of Christianized *Yijing*, see Sophie Ling-chia Wei (2020); John T. P. Lai and Wu (2019, pp. 1–17); John T. P. Lai (2021b). (Online version ahead of print).

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
