# Peer review of "Moral Cultivation and Divine Revelation: James Legge’s Religious Interpretation of the Yijing (Book of Changes)"

_religions, doi:10.3390/rel14080958_

Round 1

Reviewer 1 Report

This article offers a new perspective on James Legge's translation of the Book of Changes. Long accepted as a detached, empirical rendition of the Book of Changes as a historical account of the Western Zhou dynasty, the author argues that James Legge's translation contained deep religious meaning. This new approach allows the author to uncover Legge's understanding of Confucianism as a form of primitive monotheism -- a theme that the Figurists stressed in the 17th century. 

In my opinion, this article marks a new milestone in understanding the complexity and creativity of the Victorian translation of China. As shown in the case of James Legge, the Victorian translations were deep in theology and broad in its global reach. 

If the author wants to make the article even stronger, he or she may consider adding a brief comparison between James Legge and Thomas McClathchie, completing the discussion of the Victorian translations of the Book of Changes.

Author Response

While it's justifiable to sharply focus on James Legge's interpretation of the Yijing in this paper, I've added a line on p. 9 line 292, and a paragraph at the end of the Conclusion to briefly compare the divergent interpretations of the Yijing between James Legge and Thomas McClathchie, as well as emphasizing the importance of more in-depth comparative studies of these Victorian translations in the future research.  

Reviewer 2 Report

Interesting paper, worth  to be published. Two minor things: footnote 12, it is the Rare Book Division.

p. 5, lines 132-135: the author says two times the same thing, namely that Confucianism is not only a system of morality, but also a religion. 

Author Response

I've already made the amendment "Rare Book Division" in footnote 12, and deleted the repeated line "Confucianism is not only a system of morality, but also a religion" on p. 5, line 133-134.